# Oligo-Recurrence in Lung Cancer; The Most Curable State Among Advanced Disease?

**DOI:** 10.3390/cancers16234086

**Published:** 2024-12-06

**Authors:** Yoshihisa Shimada

**Affiliations:** Department of Thoracic Surgery, Tokyo Medical University, Tokyo 160-0023, Japan; zenkyu@za3.so-net.ne.jp

**Keywords:** oligo-recurrence, local ablative therapy, radiotherapy, surgery

## Abstract

Despite advancements in systemic therapies and precision medicine, recurrence or progression remains common in advanced non-small-cell lung cancer (NSCLC), with only a few patients achieving long-term control. For patients with oligometastases, where metastases are limited, combining systemic therapy with local ablative therapies (LATs) such as radiotherapy or surgery may offer curative potential. The concept of oligo-recurrence applies to cases where the primary lesion is controlled and all metastases are potentially treatable with LATs. Oligo-recurrence, regarded as the most favorable oligometastatic subgroup, shows promising responses to LATs, but its classification remains debated. Further research, including ongoing trials, is needed to refine strategies and optimize outcomes for this potentially curable condition.

## 1. Introduction

Lung cancer is the leading cause of cancer-related deaths globally, largely due to its aggressive nature and tendency to metastasize to other organs [1]. The extensive blood and lymphatic networks surrounding the lungs facilitate this metastatic spread, making it particularly challenging to control. In recent years, immune checkpoint inhibitors and molecular-targeted therapies have emerged as promising treatments for metastatic non-small cell lung cancer (NSCLC). However, despite their effectiveness, these therapies often show limited success in the later stages of the diseases or after recurrence, often due to acquired resistance or the presence of a more aggressive tumor biology.

For a subset of patients with more localized metastatic disease, there is growing evidence that a multimodal treatment approach, combining systemic therapy with local ablative therapies (LATs), such as radiotherapy or surgical resection, may offer curative potential. One concept that is gaining attention is oligometastases. The concept of oligometastases was proposed by Hellman and Weichselbaum in 1995 [2]. It refers to an advanced stage of cancer with a limited number of metastatic lesions and represents an intermediate disease state between locally advanced cancer and widespread systemic metastasis. More than ten years later, the concept of oligo-recurrence was defined by Niibe and colleagues, as the state in which cancer patients have five or fewer metastatic lesions with controlled primary lesions [3,4,5]. Oligo-recurrence is both cancer- and organ-specific, aligning with the “seed and soil” theory. In the case of NSCLC, oligo-recurrence frequently manifests as brain- or adrenal-only recurrences. During the treatment of the primary lesion, patients with oligo-recurrent cancer may already harbor one or more micrometastases. These micrometastases typically remain dormant for a certain period before resuming growth. Over time, they become detectable through imaging modalities. This phase, characterized by the presence of one to several macroscopic recurrences, is defined as oligo-recurrence.

In this review, I present a thorough overview of the evidence supporting the concepts of lung cancer oligometastases and oligo-recurrence. I review the efficacy of radiotherapy and surgical resection for treating metastatic lesions and, finally, discuss whether oligo-recurrence can be classified as the most curable state among advanced NSCLC cases.

## 2. Search Methods

In the narrative review, I collected international guidelines along with retrospective and prospective studies from relevant articles and reviews. The search was limited to English-language publications, and the PubMed database was queried using the terms “lung cancer” and “oligo-recurrence”.

## 3. Definition and Classification

The term “oligometastases” refer to a condition of metastatic disease that is limited in the number of metastatic sites and the extent of disease, making it potentially amenable to metastatic-directed surgical and radiotherapeutic treatments with curative intent for select patients, as outlined in an editorial written by the aforementioned two radiologists [2]. On the other hand, Niibe et al. defined the concept of oligo-recurrence in the mid-2000s as the presence of one to five metastases and recurrences (excluding local recurrence) with controlled primary lesions [5]. The key difference between oligometastases and oligo-recurrences lies in the status of the primary lesion (whether it is controlled or uncontrolled), indicating that all gross recurrent or metastatic sites can be treated with LATs.

However, it is important to note that, in past clinical trials on oligometastatic NSCLC, the definition of oligometastases varied across studies, making it difficult to compare and interpret the results. To address this issue, several consensus guidelines have been proposed, primarily by North American and European radiotherapy groups, aiming to establish standardized definitions, diagnostic criteria, and classifications for oligometastatic NSCLC. In 2019, the European Organization for Research and Treatment of Cancer Lung Cancer Group (EORTC LCG) proposed a consensus aimed at standardizing eligibility criteria in clinical trials and facilitating the comparison of clinical trial data [6]. A survey was distributed to the consensus group as well as to national societies and members of the EORTC LCG and a radiation oncology group (ROG). A total of 423 physicians from 34 countries completed the survey, and their responses helped shape the questions for the consensus meeting. According to this consensus, synchronous oligometastatic NSCLC is defined as having up to five metastases in a maximum of three organs, all of which are treatable with local therapy [6].

In 2020, the European Society for Radiotherapy and Oncology (ESTRO) and the American Society for Radiation Oncology (ASTRO) reached a consensus on the definition of oligometastatic disease, highlighting the necessity of ensuring that all metastatic lesions are amenable to safe treatment [7]. Furthermore, the 2023 edition of the Japanese Lung Cancer Society Guidelines for non-small cell lung cancer includes a section on oligometastatic disease that recommends that local therapy should be considered for patients with stage IV NSCLC who have a limited number of synchronous oligometastatic lesions and whose disease has stabilized with drug therapy [8]. In 2020, ESTRO and EORTC consensus introduced a decision tree that uses five questions to classify oligometastatic disease into nine different categories based on factors such as the presence or absence of polymetastatic disease, whether the primary tumor was diagnosed within six months, whether systemic therapy was ongoing at the time of oligometastatic disease diagnosis, and whether there were progressing lesions during systemic therapy [9]. This decision tree facilitates the development of treatments specific to each disease state and is also valuable for the efficient conducting of clinical trials. This consensus detailed oligo-recurrence, separating it into three subclassifications, namely metachronous, repeat, and induced oligo-recurrence. “Metachronous” refers to the first-time diagnosis of new oligometastases more than 6 months after the initial diagnosis of cancer. “Repeat” indicates patients with a previous diagnosis of oligometastatic disease, and the subsequent diagnosis of new, growing, or regrowing oligometastases. “Induced” refers to patients with a history of polymetastatic disease, where active systemic therapy induces oligometastases. Thus, when considering all the evidence, oligo-recurrence is currently defined as a state characterized by a limited number of metastases and recurrences with controlled primary lesions and is classified into the subtypes metachronous, repeat, and induced.

## 4. Clinical Trials for Oligometastatic and Oligo-Recurrent NSCLC

As of October 2024, no randomized clinical trial results have been published specifically for oligo-recurrent lung cancer. However, several important randomized phase II trials are underway that compare LATs that target all metastatic sites with maintenance therapy or observation for stage IV oligometastatic NSCLC. In a 2016 randomized phase II trial by Gomez et al., the effects of combining maintenance drug therapy with local treatment were evaluated in NSCLC patients with three or fewer metastatic lesions who had stable disease after initial chemotherapy for three months [10]. An interim analysis of 49 patients demonstrated a significant improvement in progression-free survival (PFS) in the local treatment group, with a hazard ratio of 0.35 (11.9 months vs. 3.9 months, 90% confidence interval 0.18–0.66, *p* = 0.006). Long-term follow-up further indicated that median overall survival (OS) in the local treatment group was 41.2 months, notably longer than the 17.0 months observed in the maintenance therapy group [11]. Additionally, adverse events were comparable between both groups. While these results are promising, the study’s small sample size and its phase II design limit the generalizability of findings. Additionally, patients were highly selected, raising concerns about applicability to broader populations.

In a 2018 single-institution randomized phase II trial by Iyengar et al., patients with NSCLC who demonstrated stable disease after 4–6 cycles of platinum-based chemotherapy, were negative for epidermal growth factor receptor (EGFR) mutations and anaplastic lymphoma kinase (ALK) fusion genes, and had up to five metastatic lesions (with no more than three in the lungs or liver) were enrolled [12]. The study compared maintenance drug therapy alone to the addition of stereotactic body radiotherapy (SBRT). An interim analysis of 29 patients showed a significant improvement in PFS in the local treatment group with a hazard ratio of 0.30 (9.7 months vs. 3.5 months, 95% confidence interval 0.11–0.82, *p* = 0.01).

In the randomized phase II SABR-COMET trial, which included patients with solid tumors and up to five metastatic lesions (including the primary tumor), the additional effect of SBRT was compared to maintenance therapy in patients whose disease had remained controlled for more than three months with drug therapy [13]. Among the participants, 18% had NSCLC. Results from the 99 randomized patients indicated a significant extension in OS in the SBRT group, with a hazard ratio of 0.57 (41 months vs. 28 months, 95% confidence interval 0.30–1.10, *p* = 0.009). However, grade 2 or higher adverse events were observed in 29% of patients (19 individuals) in the SBRT group compared to 9% (3 individuals) in the maintenance therapy group. Additionally, treatment-related deaths occurred in three patients (4.5%) in the SBRT group, while no such events were reported in the maintenance therapy group [13]. While the results indicate the potential of SBRT, the study’s heterogeneous population and non-specific tumor focus limit conclusions about NSCLC-specific efficacy. The higher adverse event rate in the SBRT group also warrants caution.

The SINDAS trial studied first-line TKI therapy with or without upfront radiotherapy in EGFR-mutated synchronous oligometastatic NSCLC [14]. Between 2016 and 2019, 133 patients with up to five extracranial metastases were enrolled. Median PFS was 12.5 months with TKI alone versus 20.2 months with TKI plus SBRT, and median OS was 17.4 months versus 25.5 months, respectively.

Currently, several phase III trials targeting lung cancer oligometastases are ongoing. While no final reports have been published, Iyengar et al. presented findings at the 2024 American Society of Clinical Oncology annual meeting from a randomized phase II/III trial (NRG-LU002) comparing maintenance therapy alone to local consolidation therapy plus maintenance therapy in oligometastatic NSCLC [15]. The phase II results showed no significant difference in PFS between the groups, leading to the study’s closure for further enrollment. This outcome may be due to the heterogeneous study population, which included both true oligometastasis cases with fewer initial metastatic sites and induced oligometastasis cases, where systemic therapy initially controlled widespread metastasis to create an oligometastatic state.

The JCOG2108 trial, led by the Japan Clinical Oncology Group (JCOG), is a unique ongoing multicenter randomized phase III study evaluating the efficacy of systemic chemotherapy followed by maintenance therapy compared to local consolidation therapy in patients with postoperative oligometastatic recurrent NSCLC [16]. This study focuses on patients with completely resected, non-EGFR/ALK-mutated NSCLC with three or fewer distant recurrent lesions, with OS acting as the primary endpoint. After initial chemo-immunotherapy, patients with reduced or stable metastases are randomized to receive either maintenance therapy or LATs (radiotherapy or surgery) targeting all recurrent lesions. The results of this study could help establish a dedicated treatment framework, including LATs, for oligo-recurrent NSCLC.

Targeted next-generation sequencing is now a routine part of clinical practice for cancer. The clinical trials discussed here, along with ongoing phase III trials on oligometastatic NSCLC, are designed to incorporate multiple molecular-targeted therapies and immune checkpoint inhibitors as systemic treatments. The advent of effective drug therapies and precision medicine has improved outcomes for patients with advanced NSCLC and even for those with oligometastatic disease. Combining systemic therapies with LATs is crucial in pursuing a potential cure for oligometastatic NSCLC.

The studies discussed emphasize the potential benefits of integrating LATs with systemic therapies for oligometastatic NSCLC. However, the heterogeneity of trial designs, patient populations, and outcomes highlights the need for further research. Combining systemic therapies with LATs holds promise as a curative strategy, but careful consideration of patient selection and treatment timing is crucial. Further trials must address these challenges to provide robust evidence for clinical practice.

## 5. Surgery for Oligo-Recurrent NSCLC

In consensus and clinical practice guidelines, curative local treatment for oligometastatic NSCLC generally refers to metastasis-directed therapy, primarily radiotherapy. Among the randomized phase II trials discussed earlier, only the study by Gomez et al. permitted surgical treatment as a form of LAT, with surgery being used in just 28% of cases [10]. Clear criteria for selecting radiotherapy versus surgery as the preferred local therapy remain undefined. Even in ongoing randomized phase III trials, surgery as an LAT option is only included in a few, notably the JCOG2108 trial, which allows both surgery and radiotherapy for oligo-recurrent NSCLC [16]. However, this does not imply that radiotherapy is inherently superior to surgery. Recent reports increasingly highlight the potential role of surgical treatment in oligometastatic lung cancer.

Mitchell et al. conducted a retrospective study on patients with NSCLC (cT1-3N0-2M1) and three or fewer synchronous metastases, comparing LAT by pulmonary resection or radiotherapy for the primary lesion [17]. Of 88 patients, 63 (71.6%) received radiotherapy while 25 (28.4%) underwent surgery (lobectomy in 80%, pneumonectomy in 12%, sublobar in 8%). Ninety-day mortality was low (0% for surgery; 1.6% for radiotherapy). Median survival was 55.2 months in the surgery group and 23.4 months in the radiotherapy group, suggesting that pulmonary resection is feasible and may offer long-term survival for selected patients, though selection bias may exist.

Deboever et al. retrospectively analyzed patients with oligometastatic NSCLC (>3 synchronous metastases) who underwent primary tumor resection. Among the 52 patients meeting the criteria, there were no deaths at 30 or 90 days [18]. Median postoperative PFS was 9.4 months (5.5–11.6) and OS was 51.7 months (22.3–65.3), indicating that pulmonary resection for locoregional control in oligometastatic NSCLC is feasible, safe, and may offer durable long-term survival benefits. While the safety profile of surgical resection was excellent, the study’s limit cohort and lack of a control group limit its generalizability. The results underscore the potential of surgical resection but fail to establish its superiority over other LAT modalities.

Regarding LAT modalities, retrospective studies on oligo-recurrent NSCLC vary in focus: many examine both surgery and radiotherapy, others focus on radiotherapy alone, and a few evaluate surgical resection alone. Ishige et al. reported on seven patients with liver oligo-recurrent NSCLC who underwent hepatectomy [19]. Complete tumor resection was achieved in all patients without complications, with a median survival of 24.0 months (range 15.2–30.2) following hepatectomy. This finding suggests that hepatectomy could be as effective as multidisciplinary approaches for managing liver oligo-recurrent NSCLC. In another study, Raz et al. examined outcomes for patients with oligometastatic adrenal metastases from NSCLC, comparing surgical and non-surgical treatments [20]. Patients who underwent surgery had a significantly higher five-year overall survival rate of 34%, whereas those receiving non-surgical treatment had a survival rate of 0% (*p* = 0.002), though the results may be influenced by selection bias.

Table 1 summarizes retrospective studies on patients with oligo-recurrent NSCLC after curative control of the primary lung tumor treated with radiotherapy and/or surgery for recurrence [21,22,23,24,25,26,27,28,29,30,31,32]. Hishida et al. analyzed 768 NSCLC patients with postoperative recurrence after complete resection [23]. The earlier the stage, the higher the prevalence of oligo-recurrence. Among the 162 patients with three or fewer metastases, the 5-year post-recurrence survival rate was 32.9%, significantly higher than in those with more than three lesions. Multivariate analysis showed that initial local therapy (radiation or surgery) for recurrent disease was associated with improved post-recurrence survival (odds ratio [OR] 0.44; 95% confidence interval [CI] 0.29–0.68). Sonoda et al. examined post-recurrence survival in 118 NSCLC patients with oligo-recurrence after complete resection, stratified by EGFR mutation status [30]. They found that local treatment of metastatic lesions significantly improved survival in patients without or with unknown EGFR mutations but not in those with EGFR mutations. These studies suggest that postoperative oligo-recurrence, accounting for 21% to 53% of all recurrences, is not uncommon. These studies highlight the importance of LATs in postoperative oligo-recurrence but reveal variations in treatment efficacy depending on genetic and clinical factors. Driver mutation status may influence treatment outcomes and should be considered in clinical decision-making for oligo-recurrent NSCLC.

When assessing postoperative oligo-recurrence in the ESTRO/EORTC consensus, it classifies as “synchronous oligometastatic disease” because it involves patients with no prior systemic treatment for recurrence (genuine oligometastatic disease), represents a first diagnosis of oligometastases (de novo oligometastatic disease), and is characterized by metachronous metastasis. These homogeneous states facilitate establishing objective criteria for choosing between surgical treatment and radiotherapy.

Surgery appears promising for selected patients but requires further validation. The integration of genetic profiling and standardized treatment criteria will be pivotal in advancing the management of oligo-recurrent NSCLC.

## 6. Radiotherapy for Oligo-Recurrent NSCLC

In 2006, Niibe et al. conducted a pivotal multi-institutional study that first proposed the concept of oligo-recurrence. The study included 84 patients with isolated oligo-recurrent para-aortic lymph nodes who were treated with conventional radiotherapy either alone or in combination with chemotherapy. The treatment resulted in a 5-year overall survival rate of 31.3% [4]. In 2008, Niibe et al. expanded their research on conventional radiotherapy by focusing on isolated osseous metastases in breast cancer. They treated seven patients with solitary bone metastases, where the primary tumor and other metastatic sites were well controlled, using radiation doses ranging from 30 to 50 Gy [33]. Only one patient, who received the lowest dose of 30 Gy, experienced a recurrence of pain. These results indicate that radiation therapy for solitary osseous metastasis can be effective, with higher doses potentially offering sustained pain relief and improved outcomes.

Retrospective studies on lung cancer oligo-recurrence are summarized in Table 2 [34,35,36,37,38,39,40]. Yamashita et al. classified oligometastases into sync-oligometastases and oligo-recurrence [35]. Sync-oligometastases refer to cases with ≤5 metastatic or recurrent lesions alongside active primary tumors. A study of 96 patients, including 10 with sync-oligometastases, 79 with oligo-recurrence, and 7 unclassified cases, reported a median follow-up of 32 months for survivors. Patients with oligo-recurrence showed a significantly longer median OS of 66.6 months compared to 23.9 months for those with sync-oligometastases (*p* = 0.0029). A multivariate analysis identified sync-oligometastases and multiple metastatic lesions as key factors associated with poorer OS and RFS, underscoring their unfavorable prognostic impact. In a separate analysis, Niibe et al. examined the prognostic impact of pulmonary oligo-recurrence versus sync-oligometastases [38]. Among 1378 patients with 1547 tumors (1016 oligo-recurrence, 118 sync-oligometastases, and 121 unclassified), the three-year OS was 64.0% for oligo-recurrence and 47.5% for sync-oligometastases (*p* < 0.001). Multivariate analysis showed a hazard ratio (HR) of 1.601 for sync-oligometastases compared to oligo-recurrence (*p* = 0.014). Collectively, these results suggest that OS is better for oligo-recurrence patients than for those with sync-oligometastases. These studies highlight the importance of distinguishing between sync-oligometastases and oligo-recurrence. Prognoses differ, emphasizing the need for personalized treatment approaches.

Kissel et al. evaluated outcomes for lung cancer patients with extracranial metastases across various oligometastatic settings—oligometastatic, oligorecurrent, oligopersistent, and oligoprogressive (the “oligometastatic spectrum”)—using SBRT (stereotactic body radiation therapy) with or without systemic treatments [41]. They found that initial nodal stage and the specific oligometastatic spectrum category were prognostic factors for distant progression-free survival (d PFS), while age, initial primary stage, and oligometastatic spectrum were prognostic factors for OS in multivariate analysis. Among the oligometastatic spectrum, oligoprogression was identified as an unfavorable prognostic factor.

Niibe et al. examined the prognostic significance of oligo-recurrence in NSCLC patients with brain-only oligometastases treated with stereotactic radiosurgery (SRS) or stereotactic radiotherapy (SRT) [36]. The median OS for all 61 patients was 26 months (95% CI: 17.5–34.5 months). When categorized by oligostatus, the sync-oligometastases group had a median OS of 18 months (95% CI: 14.8–21.1 months) with a five-year OS of 0%, while the oligo-recurrence group had a median OS of 41 months (95% CI: 27.8–54.2 months) with a five-year OS of 18.6%. In multivariate analysis, oligo-recurrence was the only significant independent factor associated with a favorable prognosis.

Aoki et al. evaluated the feasibility of salvage SBRT for patients who had undergone surgery and subsequently developed local failure or intrathoracic oligo-recurrence [37]. In total, 52 patients with 59 tumors were analyzed. The median OS following salvage SBRT was 32 months, with one- and three-year OS rates of 84.4% and 67.8%, respectively. Only four patients (7.7%) experienced local failure. The study concluded that postoperative salvage SBRT is a promising therapeutic option for NSCLC patients with local failure or intrathoracic oligo-recurrence.

Radiotherapy is essential for the local treatment of oligometastases, serving not only curative purposes but also playing a critical role in complementing and enhancing systemic therapy. In Japan, SBRT for oligometastases became covered by health insurance in April 2020. Further prospective studies or larger cohort studies are needed to establish robust evidence supporting radiotherapy as the primary modality of LAT for oligo-recurrent NSCLC.

The concept of oligo-recurrence, pioneered by Niibe et al., has significantly influenced the treatment landscape for oligometastatic NSCLC. Retrospective studies highlight the potential of radiotherapy, particularly SBRT and SRS, in improving surgical outcomes. However, the limitations of exiting evidence underscore the well-designed prospective trials. Tailoring treatment based on the specific oligometastatic category and integrating local and systemic therapies will be critical for advancing care in oligo-recurrent NSCLC.

## 7. Other Treatments for Oligo-Recurrent NSCLC

While radiotherapy and surgery remain the primary local therapeutic options for oligo-recurrent NSCLC, other modalities, such as radiofrequency ablation (RFA) and microwave ablation (MWA), are emerging as potential alternatives. Hiraki et al. reported promising results for RFA in treating NSCLC patients with pulmonary metastasis from various primary cancers, including colorectal, lung, hepatocellular carcinoma, renal cell carcinoma, and sarcoma, suggesting RFA’s potential to provide long-term survival for patients with oligo-recurrence or oligometastasis of lung cancer [42]. Regarding oligo-recurrence of NSCLC, Kodama et al. conducted a notable study involving 44 patients who underwent lung RFA for recurrent NSCLC post-surgery [43]. Of these, 43 had no extrapulmonary metastasis, while one patient with liver and splenic metastasis also received RFA. They treated single or multiple intrapulmonary recurrences, with an average follow-up of 29 months, showing overall survival rates of 98% at one year, 73% at two years, and 56% at three years. Recurrence-free survival rates were 77% at one year and 41% at three years, with tumor size and sex identified as independent significant predictors in multivariate analysis. This study suggested that RFA could offer a potential for long-term survival in patients with oligo-recurrent primary lung cancer. However, the applicability of RFA to larger or multiple tumors remains unclear, as outcomes seem dependent on tumor size. RFA offers a minimally invasive alternative for selected patients with oligo-recurrent NSCLC, particularly those unfit for surgery or radiotherapy. Further studies are needed to clarify its role in the context of evolving radiotherapy and surgical techniques.

In a retrospective study, Ni et al. evaluated 103 patients with pulmonary oligo-recurrence following curative surgical resection for NSCLC. A total of 143 MWA procedures were carried out to target and treat all recurrent nodules, highlighting the role of MWA as a therapeutic option for managing localized recurrences [44]. The study reported median PFS of 15.1 months and OS of 40.6 months. Among the patients treated with MWA, 15 (14.6%) experienced local recurrence as the first sign of disease progression, while 45 (43.7%) developed intrathoracic oligo-recurrence and 20 (19.4%) showed distant metastases. While the study demonstrates MWA’s efficacy, its retrospective design and absence of direct comparisons to other modalities limit its generalizability. Additionally, the study did not evaluate complications or quality of life.

However, advancements in radiotherapy and surgical techniques may have further improved the curability of oligo-recurrent NSCLC, potentially reducing the role of RFA or MWA in regard to oligo-recurrence today.

## 8. Conclusions and Future Directions

With advancements in both systemic and local therapies, there is increasing interest in exploring the curative potential for oligometastases and oligo-recurrence in lung cancer. Among the various oligometastatic subclassifications outlined in the ESTRO-EORTC consensus, oligo-recurrence stands out as the subgroup with the most favorable prognosis, showing significant potential for benefit from LATs. However, this classification is still subject to ongoing debate and requires further investigation. Even though oligo-recurrence may represent a favorable prognostic group among advanced diseases and is relatively homogeneous as a metastatic type, various clinical scenarios still exist depending on factors such as the number of metastatic sites, the type of affected organs, and the timing of occurrence. The choice of modality—surgery or radiotherapy—depends on the specific scenario. For example, in cases of ipsilateral lung metastasis at the site of the primary lesion that was resected, radiotherapy is often preferable due to the risk of potential hilar adhesions, which can increase surgical stress. Conversely, bilateral SBRT can pose risks, so peripheral or contralateral lung nodules may be better managed with surgical resection. Furthermore, considering that oligometastatic diseases may still have systemic characteristics, systemic therapies tailored to the patient’s specific driver mutations and programed death-ligand 1 expression should be incorporated into the treatment strategy.

Beyond the ongoing JCOG2108 trial, additional rigorous clinical trials specifically focusing on oligo-recurrence could provide valuable insights into refining strategies for treating this advanced but potentially curable state of NSCLC. These studies will be crucial in determining the most effective, tailored approaches to maximize outcomes for patients within this prognostically favorable subgroup.

## Figures and Tables

**Table 1 cancers-16-04086-t001:** Retrospective studies of lung cancer oligo-recurrence after surgical resection receiving surgery and/or radiotherapy for recurrent diseases.

Author, Published Year	No. of Patients with Oligo-Recurrence	Local Treatment Modality	No. of Metastatic Sites	The Ratio of Oligo-Recurrence	Prognosis After Oligo-Recurrence
Yano et al., 2013 [21]	17	Radiation, Surgery	1–3	33%	PFS 20 months
Shimada et al., 2015 [22]	76	Radiation, Surgery	1–5	35%	Five year OS 40.0%
Hishida et al., 2016 [23]	162	Radiation, Surgery	1–3	21%	Five year OS 32.9%
Yuan et al., 2020 [24]	56	Radiation, Surgery	1–3	10.3%	OS 31 months
Matsuguma et al., 2020 [25]	217	Radiation, Surgery	1–3	53%	Five year OS 20.7%
Han et al., 2020 [26]	102	Radiation, Surgery	1–5	21.0%	OS 46.4 months
Sonoda et al., 2021 [27]	97	Radiation, Surgery	1–5	18.1%	Five year OS 61.5% (DLT group)
Sonoda et al., 2022 [28]	214	Radiation, Surgery	1–2	37%	Five year OS 33.6%
Jin et al., 2022 [29]	81	Radiation, Surgery	1–3	-	Five year OS 26.5% (DLT group)
Sonoda et al., 2023 [30]	118	Radiation, Surgery	1–2	48%	Five year OS 59.4% (EGFRm + DLT)
Sonoda et al., 2024 [32]	125	Radiation, Surgery	1–2	46.8%	Five year OS 42.8%
Tachibana et al., 2024 [31]	66	Radiation, Surgery	1–3	23%	Unknown

PFS, post-recurrence progression-free survival. OS, post-recurrence overall survival. DLT, definitive local therapy. EGFRm, epidermal growth factor receptor mutation positive.

**Table 2 cancers-16-04086-t002:** Retrospective studies of lung cancer oligo-recurrence after surgical resection receiving radiotherapy for recurrent diseases.

Author, Published Year	No. of Patients with Oligo-Recurrence	Local Treatment Modality	No. of Metastatic Sites	Prognosis After Oligo-Recurrence
Takahashi et al., 2012 [34]	42	SBRT	1–3	2 year OS 65%
Yamashita et al., 2016 [35]	79	SBRT	1–5	4 year OS 54.0%
Niibe et al., 2016 [36]	61	SRS, SRT	1–4	OS 26 months
Aoki et al., 2020 [37]	217	SBRT	-	3 year OS 67.8%
Niibe et al. 2020 [38]	1016	SBRT	1–5	3 year OS 64.0%
Lin et al., 2022 [39]	50	SBRT	1–3	3 year OS 45.3%
Ebadi et al., 2023 [40]	168	SBRT	1–5	OS 31.3 months

OS, post-recurrence overall survival.

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
