# Peer review of "Oligo-Recurrence in Lung Cancer; The Most Curable State Among Advanced Disease?"

_cancers, 2024, doi:10.3390/cancers16234086_

Round 1
Reviewer 1 Report
Comments and Suggestions for Authors
1) The abstract must be rewritten. The current abstract only defines and describes NSCLC and local ablative therapies. The authors should revise it by providing background in 1-2 sentences, describing the aims of the review in 1 sentence, and describing the most important results from previous works. Numerical results are preferred.
2) I strongly recommend adding a graphical abstract at the end of the introduction.
3) Is there any relationship between lung cancer’s stages (I-IV) and oligo-recurrence?
4) It would be very interesting if the authors provided a table of oligo-recurrence in different countries or continents.
5) It is good the authors pay attention to mechanisms behind the oligo-recurrence.
6) In line 148, the authors mentioned “ several phase III trials targeting lung cancer oligo metastases are ongoing”. I would like to ask whether these drug candidates are specific to oligo-recurrence or normal anticancer drugs to prevent cancer cell proliferation.
Author Response
November 30, 2024
Dr. Samuel C. Mok
Editor-in-Chief
Manuscript ID cancers-3314797“Oligo-recurrence in Lung Cancer; The Most Curable State Among Advanced Disease?”
Dear Mok, MD
Thank you so much for your letter of 22-November-2024, stating your comments to my manuscript. I have carefully revised my manuscript and responded all the comments from reviewers as follows.
Reviewer #1
Comments and Suggestions for Authors
- The abstract must be rewritten. The current abstract only defines and describes NSCLC and local ablative therapies. The authors should revise it by providing background in 1-2 sentences, describing the aims of the review in 1 sentence, and describing the most important results from previous works. Numerical results are preferred.
Answer to 1) Thank you for your comment. According to this, I revised the Abstract as follows;
Change-
Despite the introduction of effective systemic therapies and advancements in precision medicine, recurrence or progression remains common in advanced non-small cell lung cancer (NSCLC). For a subset of patients with more localized metastatic disease—referred to as oligometastases and oligo-recurrence—emerging evidence suggests that a multimodal approach combining systemic therapy with local ablative therapies (LAT) may offer curative potential. The oligo-recurrence is defined by the presence of a limited number of metastases and recurrences in patients with controlled primary lesions. In this review, we focus on providing a comprehensive overview of the evidence supporting the concepts of oligo-recurrence in lung cancer, which is considered one of the most curable states among advanced diseases. Although the definition remains variable and is still under discussion, retrospective studies have reported that it is not a rare condition (occurring in 18-53% of cases) and shows relatively better survival outcomes, regardless of whether local ablative therapy (LAT) is performed. However, this classification remains a topic of ongoing debate and warrants further exploration. In addition to an ongoing randomized clinical trial on oligo-recurrent NSCLC, further rigorous studies specifically addressing oligo-recurrence are needed to refine treatment strategies for this advanced yet potentially curable state. These investigations are essential for developing effective, tailored approaches to optimize outcomes for patients within this prognostically favorable subgroup. (Page 1)
- I strongly recommend adding a graphical abstract at the end of the introduction.
Answer to 2) Thank you for this comment. I added a graphical abstract as follows.
Change-
- Is there any relationship between lung cancer’s stages (I-IV) and oligo-recurrence?
Answer to 3) Thank you for your comment. Yes, lung cancer staging has been reported to be associated with the incidence of oligo-recurrence. The earlier the stage, the higher the prevalence of oligo-recurrence. I have added the following phrase.
Change-
The earlier the stage, the higher the prevalence of oligo-recurrence. (Page 8)
- It would be very interesting if the authors provided a table of oligo-recurrence in different countries or continents.
Answer to 4) Thank you for your comment. Unfortunately, there are only a few retrospective studies on oligo-recurrent NSCLC. The studies I included in the table cover the majority of the available research. Therefore, I am unable to provide a table that compares oligo-recurrence across different countries and continents.
- It is good the authors pay attention to mechanisms behind the oligo-recurrence.
Answer to 5) Thank you for this comment. I added the following phrase in the Introduction session.
Change-
Oligo-recurrence is both cancer- and organ-specific, aligning with the "seed and soil" theory. In the case of NSCLC, oligo-recurrence frequently manifests as brain or adrenal-only recurrences. During the treatment of the primary lesion, patients with oligo-recurrent cancer may already harbor one or more micrometastases. These micrometastases typically remain dormant for a certain period before resuming growth. Over time, they become detectable through imaging modalities. This phase, characterized by the presence of one to several macroscopic recurrences, is defined as oligo-recurrence. (Page 4)
- In line 148, the authors mentioned “several phase III trials targeting lung cancer oligo metastases are ongoing”. I would like to ask whether these drug candidates are specific to oligo-recurrence or normal anticancer drugs to prevent cancer cell proliferation.
Answer to 6) Thank you for your comment. To date, there has been only one phase III study specifically addressing lung cancer oligo-recurrence. Other studies primarily focus on de novo oligometastasis or induced oligometastasis, which refers to previously diffuse metastases that have been controlled. This highlights a significant challenge in conducting clinical studies on oligometastasis: the inclusion of heterogeneous forms of oligometastasis under a single framework, despite their differences.
The comments offered by the reviewers have been helpful in formulating what I believe is a stronger paper. I appreciate these thoughtful comments, and hope that my manuscript is now suitable for publication in Cancers.
All related correspondence should be sent to Yoshihisa Shimada, M.D., Ph.D.
Department of Surgery, Tokyo Medical University Hospital
6-7-1 Nishishinjuku, Shinjyuku-ku, Tokyo, 160-0023, Japan
Phone: +81-(0)3-3342-6111, Fax: +81-(0)3-3342-6203
E-male: zenkyu@za3.so-net.ne.jp
Yoshihisa Shimada, M.D., Ph.D.
Reviewer 2 Report
Comments and Suggestions for Authors
The manuscript, "Oligo-recurrence in Lung Cancer: The Most Curable State Among Advanced Disease?" is a narrative review of the concept of oligo-recurrence in lung cancer that highlights the potential curative approach for these patients through a combination of systemic therapies and local ablative therapies (LAT) like radiotherapy or surgery. It also covers various clinical trials and studies that support the efficacy of LAT in improving survival outcomes for patients with oligo-recurrent non-small cell lung cancer (NSCLC).
I found this review is comprehensive, focuses on curative potential and relies on an evidence-based approach from various clinical trials and studies (SABR-COMET and SINDAS trials). It highlights the role of advancements in precision medicine, such as immune checkpoint inhibitors and molecular-targeted agents.
I can say that this review is a valuable resource for understanding and advancing the treatment of oligo-recurrent lung cancer.
Author Response
November 30, 2024
Dr. Samuel C. Mok
Editor-in-Chief
Manuscript ID cancers-3314797“Oligo-recurrence in Lung Cancer; The Most Curable State Among Advanced Disease?”
Dear Mok, MD
Thank you so much for your letter of 22-November-2024, stating your comments to my manuscript. I have carefully revised my manuscript and responded all the comments from reviewers as follows.
Reviewer #1
Comments and Suggestions for Authors
- The abstract must be rewritten. The current abstract only defines and describes NSCLC and local ablative therapies. The authors should revise it by providing background in 1-2 sentences, describing the aims of the review in 1 sentence, and describing the most important results from previous works. Numerical results are preferred.
Answer to 1) Thank you for your comment. According to this, I revised the Abstract as follows;
Change-
Despite the introduction of effective systemic therapies and advancements in precision medicine, recurrence or progression remains common in advanced non-small cell lung cancer (NSCLC). For a subset of patients with more localized metastatic disease—referred to as oligometastases and oligo-recurrence—emerging evidence suggests that a multimodal approach combining systemic therapy with local ablative therapies (LAT) may offer curative potential. The oligo-recurrence is defined by the presence of a limited number of metastases and recurrences in patients with controlled primary lesions. In this review, we focus on providing a comprehensive overview of the evidence supporting the concepts of oligo-recurrence in lung cancer, which is considered one of the most curable states among advanced diseases. Although the definition remains variable and is still under discussion, retrospective studies have reported that it is not a rare condition (occurring in 18-53% of cases) and shows relatively better survival outcomes, regardless of whether local ablative therapy (LAT) is performed. However, this classification remains a topic of ongoing debate and warrants further exploration. In addition to an ongoing randomized clinical trial on oligo-recurrent NSCLC, further rigorous studies specifically addressing oligo-recurrence are needed to refine treatment strategies for this advanced yet potentially curable state. These investigations are essential for developing effective, tailored approaches to optimize outcomes for patients within this prognostically favorable subgroup. (Page 1)
- I strongly recommend adding a graphical abstract at the end of the introduction.
Answer to 2) Thank you for this comment. I added a graphical abstract as follows.
Change-
- Is there any relationship between lung cancer’s stages (I-IV) and oligo-recurrence?
Answer to 3) Thank you for your comment. Yes, lung cancer staging has been reported to be associated with the incidence of oligo-recurrence. The earlier the stage, the higher the prevalence of oligo-recurrence. I have added the following phrase.
Change-
The earlier the stage, the higher the prevalence of oligo-recurrence. (Page 8)
- It would be very interesting if the authors provided a table of oligo-recurrence in different countries or continents.
Answer to 4) Thank you for your comment. Unfortunately, there are only a few retrospective studies on oligo-recurrent NSCLC. The studies I included in the table cover the majority of the available research. Therefore, I am unable to provide a table that compares oligo-recurrence across different countries and continents.
- It is good the authors pay attention to mechanisms behind the oligo-recurrence.
Answer to 5) Thank you for this comment. I added the following phrase in the Introduction session.
Change-
Oligo-recurrence is both cancer- and organ-specific, aligning with the "seed and soil" theory. In the case of NSCLC, oligo-recurrence frequently manifests as brain or adrenal-only recurrences. During the treatment of the primary lesion, patients with oligo-recurrent cancer may already harbor one or more micrometastases. These micrometastases typically remain dormant for a certain period before resuming growth. Over time, they become detectable through imaging modalities. This phase, characterized by the presence of one to several macroscopic recurrences, is defined as oligo-recurrence. (Page 4)
- In line 148, the authors mentioned “several phase III trials targeting lung cancer oligo metastases are ongoing”. I would like to ask whether these drug candidates are specific to oligo-recurrence or normal anticancer drugs to prevent cancer cell proliferation.
Answer to 6) Thank you for your comment. To date, there has been only one phase III study specifically addressing lung cancer oligo-recurrence. Other studies primarily focus on de novo oligometastasis or induced oligometastasis, which refers to previously diffuse metastases that have been controlled. This highlights a significant challenge in conducting clinical studies on oligometastasis: the inclusion of heterogeneous forms of oligometastasis under a single framework, despite their differences.
Reviewer #3
Comments:
- The provided sections read like narration for the evidence of discussed points without critical aspects/reflection points. A critical examination of these studies' quality and limitations could provide greater insight. Comparing prospective and retrospective data would enhance the discussion. A take-home message is also recommended at the end of each section.
Answer to 1) Thank you for this comment. According to this, I revised all through the manuscript. Here is an example.
Change-
Mitchell et al. conducted a retrospective study on patients with NSCLC (cT1-3N0-2M1) and three or fewer synchronous metastases, comparing LAT by pulmonary resection or radiotherapy for the primary lesion [17]. Of 88 patients, 63 (71.6%) received radiotherapy, while 25 (28.4%) underwent surgery (lobectomy in 80%, pneumonectomy in 12%, sublobar in 8%). Ninety-day mortality was low (0% for surgery; 1.6% for radiotherapy). Median survival was 55.2 months in the surgery group and 23.4 months in the radiotherapy group, suggesting that pulmonary resection is feasible and may offer long-term survival for selected patients, though selection bias may exist.
Deboever et al. retrospectively analyzed patients with oligometastatic NSCLC (>3 synchronous metastases) who underwent primary tumor resection. Among the 52 patients meeting the criteria, there were no deaths at 30 or 90 days [18]. Median postoperative PFS was 9.4 months (5.5–11.6), and OS was 51.7 months (22.3–65.3), indicating that pulmonary resection for locoregional control in oligometastatic NSCLC is feasible, safe, and may offer durable long-term survival benefits. While the safety profile of surgical resection was excellent, the study’s limit cohort and lack of a control group limit its generalizability. The results underscore the potential of surgical resection but fail to establish its superiority over other LAT modalities. (Page 7)
When assessing postoperative oligo-recurrence in the ESTRO/EORTC consensus, it classifies as "synchronous oligometastatic disease" because it involves patients with no prior systemic treatment for recurrence (genuine oligometastatic disease), represents a first diagnosis of oligometastases (de novo oligometastatic disease), and is characterized by metachronous metastasis. These homogeneous states facilitate establishing objective criteria for choosing between surgical treatment and radiotherapy. Surgery appears promising for selected patients but requires further validation. The integration of genetic profiling and standardized treatment criteria will be pivotal in advancing the management of oligo-recurrent NSCLC. (Page 8)
- The author is advised to highlight specific research gaps, such as the molecular characteristics that predict oligo-recurrence outcomes. Targeted therapies could be advanced with this approach.
Answer to 2) Thank you for this comment. According to this, I added the paragraph as follows.
Change-
Oligo-recurrence is both cancer- and organ-specific, aligning with the "seed and soil" theory. In the case of NSCLC, oligo-recurrence frequently manifests as brain or adrenal-only recurrences. During the treatment of the primary lesion, patients with oligo-recurrent cancer may already harbor one or more micrometastases. These micrometastases typically remain dormant for a certain period before resuming growth. Over time, they become detectable through imaging modalities. This phase, characterized by the presence of one to several macroscopic recurrences, is defined as oligo-recurrence. (Page 4)
- In order to attract the interest of more readers regarding the current review, the author is advised to summarize the treatment options for oligo-recurrent NSCLC in a colored figure, including surgery, radiotherapy, and other treatments. This point needs to be carefully addressed.
Answer to 3) Thank you for this comment. According to this, I added the graphical abstract.
Change-
- The author is advised to make the table captions stand-alone. To this end, author is advised to provide a list of abbreviations describing the full names of all the listed abbreviations in the table. This includes abbreviations such as “SBRT, SRS, and SRT” in Table 2.
Answer to 4) Thank you for this comment. I added the abbreviation.
- In lines 55-56, the author states “In the narrative review, we collected international guidelines along with retrospective and prospective studies from relevant articles and review”. In fact, the review manuscript is provided by a single author. Thus, the author may use “I” instead of we, or, better to modify the previous statement to “In the narrative review, international guidelines were collected along with retrospective and prospective studies from relevant articles and review”.
Answer to 5) Thank you for pointing out this. I revised this as follows.
Change-
In the narrative review, I collected international guidelines along with retrospective and prospective studies from relevant articles and review. The search was limited to English-language publications, and the PubMed database was queried using the terms “lung cancer” and “oligo-recurrence” (Page 4)
- The section titled "Conclusion" should be renamed "Conclusions and Future Directions" to reflect both the summary and prospective aspects of the study.
Answer to 6) Thank you for this comment. According to this, I revised it as follows.
Change-
- Conclusion and future directions (Page 12)
- In the conclusion and future directions section, the author should discuss which LAT modalities or systemic therapy combinations are most promising in terms of clinical outcomes.
Answer to 7) Thank you for this comment. According to this, I added the following phrase.
Change-
Even though oligo-recurrence may represent a favorable prognostic group among advanced diseases and is relatively homogeneous as a metastatic type, various clinical scenarios still exist depending on factors such as the number of metastatic sites, the type of affected organs, and the timing of occurrence. The choice of modality—surgery or radiotherapy—depends on the specific scenario. For example, in cases of ipsilateral lung metastasis at the site of the primary lesion that was resected, radiotherapy is often preferable due to the risk of potential hilar adhesions, which can increase surgical stress. Conversely, bilateral SBRT can pose risks, so peripheral or contralateral lung nodules may be better managed with surgical resection. Furthermore, considering that oligometastatic diseases may still have systemic characteristics, systemic therapies tailored to the patient’s specific driver mutations and programed death-ligand 1 expression should be incorporated into the treatment strategy. (Page 12)
The comments offered by the reviewers have been helpful in formulating what I believe is a stronger paper. I appreciate these thoughtful comments, and hope that my manuscript is now suitable for publication in Cancers.
All related correspondence should be sent to Yoshihisa Shimada, M.D., Ph.D.
Department of Surgery, Tokyo Medical University Hospital
6-7-1 Nishishinjuku, Shinjyuku-ku, Tokyo, 160-0023, Japan
Phone: +81-(0)3-3342-6111, Fax: +81-(0)3-3342-6203
E-male: zenkyu@za3.so-net.ne.jp
Yoshihisa Shimada, M.D., Ph.D.
Reviewer 3 Report
Comments and Suggestions for Authors
In the review entitled “Oligo-recurrence in Lung Cancer; The Most Curable State Among Advanced Disease?”, the author provided a literature review and analysis of non-small-cell lung cancer (NSCLC). Despite advancements in systemic therapies and precision medicine, recurrences of advanced NSCLC are common. In cases of oligometastases, a localized metastatic state, combined systemic and local ablative therapies (LAT) may be beneficial. An oligo-recurrence, defined as a limited number of metastases and controlled primary lesions, has a good response rate to LAT and is considered favorable. Yet, additional research is warranted to refine treatment strategies and optimize outcomes. In the current review, the author provided two interesting tables summarizing the retrospective studies of lung cancer oligo-recurrence after surgical resection receiving surgery and/or radiotherapy for recurrent diseases (Table 1) and retrospective studies of lung cancer oligo-recurrence after surgical resection receiving radiotherapy for recurrent diseases (Table 2). The summarized information is interesting, and the review is clearly written.
Comments:
1) The provided sections read like narration for the evidence of discussed points without critical aspects/reflection points. A critical examination of these studies' quality and limitations could provide greater insight. Comparing prospective and retrospective data would enhance the discussion. A take-home message is also recommended at the end of each section.
2) The author is advised to highlight specific research gaps, such as the molecular characteristics that predict oligo-recurrence outcomes. Targeted therapies could be advanced with this approach.
3) In order to attract the interest of more readers regarding the current review, the author is advised to summarize the treatment options for oligo-recurrent NSCLC in a colored figure, including surgery, radiotherapy, and other treatments. This point needs to be carefully addressed.
4) The author is advised to make the table captions stand-alone. To this end, author is advised to provide a list of abbreviations describing the full names of all the listed abbreviations in the table. This includes abbreviations such as “SBRT, SRS, and SRT” in Table 2.
5) In lines 55-56, the author states “In the narrative review, we collected international guidelines along with retrospective and prospective studies from relevant articles and review”. In fact, the review manuscript is provided by a single author. Thus, the author may use “I” instead of we, or, better to modify the previous statement to “In the narrative review, international guidelines were collected along with retrospective and prospective studies from relevant articles and review”.
6) The section titled "Conclusion" should be renamed "Conclusions and Future Directions" to reflect both the summary and prospective aspects of the study.
7) In the conclusion and future directions section, the author should discuss which LAT modalities or systemic therapy combinations are most promising in terms of clinical outcomes.
Author Response
November 30, 2024
Dr. Samuel C. Mok
Editor-in-Chief
Manuscript ID cancers-3314797“Oligo-recurrence in Lung Cancer; The Most Curable State Among Advanced Disease?”
Dear Mok, MD
Thank you so much for your letter of 22-November-2024, stating your comments to my manuscript. I have carefully revised my manuscript and responded all the comments from reviewers as follows.
Reviewer #3
Comments:
- The provided sections read like narration for the evidence of discussed points without critical aspects/reflection points. A critical examination of these studies' quality and limitations could provide greater insight. Comparing prospective and retrospective data would enhance the discussion. A take-home message is also recommended at the end of each section.
Answer to 1) Thank you for this comment. According to this, I revised all through the manuscript. Here is an example.
Change-
Mitchell et al. conducted a retrospective study on patients with NSCLC (cT1-3N0-2M1) and three or fewer synchronous metastases, comparing LAT by pulmonary resection or radiotherapy for the primary lesion [17]. Of 88 patients, 63 (71.6%) received radiotherapy, while 25 (28.4%) underwent surgery (lobectomy in 80%, pneumonectomy in 12%, sublobar in 8%). Ninety-day mortality was low (0% for surgery; 1.6% for radiotherapy). Median survival was 55.2 months in the surgery group and 23.4 months in the radiotherapy group, suggesting that pulmonary resection is feasible and may offer long-term survival for selected patients, though selection bias may exist.
Deboever et al. retrospectively analyzed patients with oligometastatic NSCLC (>3 synchronous metastases) who underwent primary tumor resection. Among the 52 patients meeting the criteria, there were no deaths at 30 or 90 days [18]. Median postoperative PFS was 9.4 months (5.5–11.6), and OS was 51.7 months (22.3–65.3), indicating that pulmonary resection for locoregional control in oligometastatic NSCLC is feasible, safe, and may offer durable long-term survival benefits. While the safety profile of surgical resection was excellent, the study’s limit cohort and lack of a control group limit its generalizability. The results underscore the potential of surgical resection but fail to establish its superiority over other LAT modalities. (Page 7)
When assessing postoperative oligo-recurrence in the ESTRO/EORTC consensus, it classifies as "synchronous oligometastatic disease" because it involves patients with no prior systemic treatment for recurrence (genuine oligometastatic disease), represents a first diagnosis of oligometastases (de novo oligometastatic disease), and is characterized by metachronous metastasis. These homogeneous states facilitate establishing objective criteria for choosing between surgical treatment and radiotherapy. Surgery appears promising for selected patients but requires further validation. The integration of genetic profiling and standardized treatment criteria will be pivotal in advancing the management of oligo-recurrent NSCLC. (Page 8)
- The author is advised to highlight specific research gaps, such as the molecular characteristics that predict oligo-recurrence outcomes. Targeted therapies could be advanced with this approach.
Answer to 2) Thank you for this comment. According to this, I added the paragraph as follows.
Change-
Oligo-recurrence is both cancer- and organ-specific, aligning with the "seed and soil" theory. In the case of NSCLC, oligo-recurrence frequently manifests as brain or adrenal-only recurrences. During the treatment of the primary lesion, patients with oligo-recurrent cancer may already harbor one or more micrometastases. These micrometastases typically remain dormant for a certain period before resuming growth. Over time, they become detectable through imaging modalities. This phase, characterized by the presence of one to several macroscopic recurrences, is defined as oligo-recurrence. (Page 4)
- In order to attract the interest of more readers regarding the current review, the author is advised to summarize the treatment options for oligo-recurrent NSCLC in a colored figure, including surgery, radiotherapy, and other treatments. This point needs to be carefully addressed.
Answer to 3) Thank you for this comment. According to this, I added the graphical abstract.
Change-
- The author is advised to make the table captions stand-alone. To this end, author is advised to provide a list of abbreviations describing the full names of all the listed abbreviations in the table. This includes abbreviations such as “SBRT, SRS, and SRT” in Table 2.
Answer to 4) Thank you for this comment. I added the abbreviation.
- In lines 55-56, the author states “In the narrative review, we collected international guidelines along with retrospective and prospective studies from relevant articles and review”. In fact, the review manuscript is provided by a single author. Thus, the author may use “I” instead of we, or, better to modify the previous statement to “In the narrative review, international guidelines were collected along with retrospective and prospective studies from relevant articles and review”.
Answer to 5) Thank you for pointing out this. I revised this as follows.
Change-
In the narrative review, I collected international guidelines along with retrospective and prospective studies from relevant articles and review. The search was limited to English-language publications, and the PubMed database was queried using the terms “lung cancer” and “oligo-recurrence” (Page 4)
- The section titled "Conclusion" should be renamed "Conclusions and Future Directions" to reflect both the summary and prospective aspects of the study.
Answer to 6) Thank you for this comment. According to this, I revised it as follows.
Change-
- Conclusion and future directions (Page 12)
- In the conclusion and future directions section, the author should discuss which LAT modalities or systemic therapy combinations are most promising in terms of clinical outcomes.
Answer to 7) Thank you for this comment. According to this, I added the following phrase.
Change-
Even though oligo-recurrence may represent a favorable prognostic group among advanced diseases and is relatively homogeneous as a metastatic type, various clinical scenarios still exist depending on factors such as the number of metastatic sites, the type of affected organs, and the timing of occurrence. The choice of modality—surgery or radiotherapy—depends on the specific scenario. For example, in cases of ipsilateral lung metastasis at the site of the primary lesion that was resected, radiotherapy is often preferable due to the risk of potential hilar adhesions, which can increase surgical stress. Conversely, bilateral SBRT can pose risks, so peripheral or contralateral lung nodules may be better managed with surgical resection. Furthermore, considering that oligometastatic diseases may still have systemic characteristics, systemic therapies tailored to the patient’s specific driver mutations and programed death-ligand 1 expression should be incorporated into the treatment strategy. (Page 12)
The comments offered by the reviewers have been helpful in formulating what I believe is a stronger paper. I appreciate these thoughtful comments, and hope that my manuscript is now suitable for publication in Cancers.
All related correspondence should be sent to Yoshihisa Shimada, M.D., Ph.D.
Department of Surgery, Tokyo Medical University Hospital
6-7-1 Nishishinjuku, Shinjyuku-ku, Tokyo, 160-0023, Japan
Phone: +81-(0)3-3342-6111, Fax: +81-(0)3-3342-6203
E-male: zenkyu@za3.so-net.ne.jp
Yoshihisa Shimada, M.D., Ph.D.
Round 2
Reviewer 1 Report
Comments and Suggestions for Authors
The manuscript has been improved and I recommend publishing it.
Reviewer 3 Report
Comments and Suggestions for Authors
The authors have adequately addressed the comments raised. Thanks.